



# Characteristics and controlling factors of the drought runoff coefficient

Rei Itsukushima[1]

[1]Department of Transdisciplinary Science and Engineering, Tokyo Institute of Technology, 4259 G5-4 Nagatsuta-cho, Midori-ku, Yokohama 226-8502, Japan

*Correspondence to*: Rei Itsukushima (itsukushima.r.aa@m.titech.ac.jp)

**Abstract.** Increasing water demand due to population growth and economic development or changes in rainfall pattern as a result of climate change is likely to alter the duration and magnitude of droughts. To establish sustainable water resource management based on changes in future drought risk, understanding the relationship between low-flow conditions and controlling factors relative to drought magnitude is important. This study is the first attempt at revealing the relationship between low-flow and controlling factors at differing drought severities. I calculated the drought runoff coefficient for six types of occurrence probability based on past observation data of minimum flow and precipitation. Furthermore, I investigated the pattern of change in the drought runoff coefficient in accordance with the occurrence probability and relationship between the coefficient and geological, land use, and topographical factors. The drought runoff coefficient for multiple drought magnitudes exhibited three behaviour types corresponding to precipitation pattern. The results from a generalized linear model (GLM) revealed that the controlling factors differ depending on drought magnitude. In high-frequency drought, the drought runoff coefficient was influenced by geological and vegetation factors, whereas land use and topographical factors influenced the drought runoff coefficient in low-frequency drought. These differences were caused by differences in the runoff component, which dominates stream discharge according to drought magnitude. Therefore, for effective water resource management, estimation of the drought runoff volume needs to consider precipitation pattern, geology, land use, and topography.

## 1 Introduction

The causes of, and adaptations to, droughts as natural disaster have been researched from the perspective of hydrology, environmentology, geology, meteorology, and agronomy (Mishra & Singh, 2010). The causes of droughts have been investigated in various regions, by focusing on rainfall pattern (Verschuren et al., 2000; Tabari, et al., 2012; Tfwala et al., 2018), temperature (Nicholls, 2004; Hein et al., 2019), wind (Namias, 1989), and humidity (Behrangi et al., 2015). In addition to the impacts of natural factors, aggravation of the drought hazard is expected to occur because of growing water demand associated with population growth and economic development (Frederiksen, 1996; Xiao-jun et al., 2012; El Kharraz et al., 2012) and changes in the hydrological cycle associated with anthropogenic impacts such as land use change (Liu et al., 2017; Deo et al., 2009; Lee et al., 2011).





Droughts are generally categorized into four types (Wilhite & Glantz, 1985). First, drought resulting from precipitation shortage is defined as meteorological drought (Mishra & Singh, 2010; Smakhtin & Hughes, 2007). In addition, the relationship between monthly rainfall and cumulative precipitation amount was investigated (Eltahir, 1992). Second, a shortage of surface or subsurface water in relation to water utilization, as determined by established water resource management, is defined as hydrological drought (Tallaksen & Van Lanen, 2004; Nalbantis & Tsakiris, 2009). Stream water discharge is often used as an

indicator of the management and analysis of hydrological drought (Clausen & Pearson, 1995). Third, agricultural drought indicates declining soil moisture, regardless of surface water resources, causing crop failure (Rickard, 1960; Nieuwolt, 1986). Finally, socio-economic drought occurs in cases of defectiveness and incompatibility of the water resource system in relation to water demand (Eklund & Seaquist, 2015; Mehran et al., 2015).

      Prolonged droughts cause severe socio-economic loss (Carrão et al., 2018; Ahmadalipour et al., 2019). Research results

on the evaluation of the economic loss of droughts indicate that damage of \$6–8 billion per year occurs in the United States (Smith & Katz, 2013; Smith & Matthews, 2015), with the EU suffering damages of €100 billion over the last 30 years (Carrão et al., 2016). The human damage caused by drought is even more serious. Droughts in Ethiopia/Sudan (1984) and the Sahel region (1974) killed 450,000 and 325,000 people, respectively (Vicente-Serrano et al., 2012).

      In addition, changes in the hydrological cycle as a result of climate change are expected to increase extreme drought events

(Mishra & Singh, 2010). Unlike flood disasters, the influence of climate change on drought is not yet fully understood. However, future prediction of drought aggravation due to population growth in central Africa (Ahmadalipour et al., 2019) and increasing drought duration and severity in the interior southwest of the United States (Andreadis & Lettenmaier, 2006) have been reported. Furthermore, forecasts of drought using soil moisture as an indicator have indicated increasingly frequent drought events in Europe regardless of emission scenario (Grillakis, 2019).

Stream flow discharge is an important indicator of hydrological drought because in many regions water resources are obtained from surface water. Previous studies of stream discharge have focused on water resources, ecosystems, river channel formation, and flood management. In particular, the effects of flow regime alteration on ecosystems have been studied (Sparks, 1995; Bunn & Arthington, 2002; Taylor et al., 2008), and the natural flow regime has been elucidated (Poff et al., 1997; Lytle & Poff, 2004; Naiman et al., 2008; Kennard et al., 2010). Research on the factors that influence flow discharge have focused

on rainfall amount or pattern (Obled et al., 1994; Montgomery et al., 1997), land use (Kashaigili, 2008; McIntyre & Marshall, 2010), and watershed geology (Meijerink, 1985). For research on flow regime, the factors influencing low flows strongly related to drought have been investigated, through focusing on watershed area, watershed elevation, ratio of urban area or forest cover, and geology (Mushiake et al., 1981; Zecharias & Brutsaert, 1988; Vogel & Kroll, 1992). However, these studies mainly focused on mountainous watersheds or a single factor. In addition, the low flows prevalent in the above research were

not probabilistically evaluated. Therefore, the relationship between the appearance frequency of low flow and its controlling factors remain unknown.

      Increasing water demand due to population growth and economic development and/or changes in rainfall pattern due to climate change alter the duration and magnitude of droughts. To establish sustainable water resource management based on





changes in future drought risk, understanding the relationship between low flow and its controlling factors in relation to the
magnitude of drought is important. Based on the above, I formulated the hypothesis that the controlling factors of low surface
flow vary according to drought severity. The present study is a first attempt at revealing this relationship. The surface water
volume of each drought occurrence probability was calculated based on long-term observation data. The relationship between
the drought water volume of each occurrence probability and the controlling factors was analyzed. Multiple controlling factors
related to geology, land use, and topography were introduced. Since the research results identify the controlling factor of
drought for each occurrence probability, they may contribute to the development of effective water resource management
through prediction of drought water volumes or impacts of climate change on the surface water runoff.

## 2 Materials and Methods

### 2.1 Location of the study area

In this study, 44 watersheds belonging to the Japanese archipelago where discharge observations have been conducted over
30 years were used. To extract stations where the impact of flow regime regulation due to a dam is small, observation stations
whose watershed was subject to over 10% occupancy by a dam watershed were excluded (Fig. 1). The watershed areas ranged
from 47 to 8,208 km2.

### 2.2 Calculation of the hydrological data

Annual total discharge of each watershed was obtained from the Water Information System (http://www1.river.go.jp/). A
sample of annual total discharge of each observation point was statistically calculated to estimate the total discharge for
occurrence probabilities of 2, 10, 30, 50, 100, and 400 years. Hydrological Statistics Utility (ver. 1.5.) was used for the
statistical analysis. I calculated the estimated design magnitude using 13 probability distributions including the exponential
distribution (EXP), Gumbel distribution (Gumbel), exponential-type distribution of maximum (SqrtEt), generalized extreme
value distribution (Gev), log-Pearson type III distribution (real coordinate space) (LP3Rs), log-Pearson type III distribution
(log coordinate space) (LogP3), Iwai method (Iwai), Ishihara Takase method (IshiTaka), the logarithmic normal distribution
with three parameters (quantile method) (LN3Q), the logarithmic normal distribution with three parameters (Slade II)
(LN3PM), the logarithmic normal distribution with two parameters (Slade I, L-moments method) (LN2LM), the logarithmic
normal distribution with two parameters (Slade I, moments method) (LN2PM), and the logarithmic normal distribution with
four parameters (Slade IV, moments method) (LN4PM). Among the 13 probability distributions, the estimated design
magnitude was selected based on standard least squares criteria (Takasao et al., 1986).

Annual precipitation data were obtained from the database of the Japan Meteorological Agency
(http://www.jma.go.jp/jma/index.html). Data from observation stations with an observation period of over 30 years were used.
A sample of the average depth of rainfall over the watershed area was calculated using a Voronoi diagram. The estimated





annual precipitation for occurrence probabilities of 2, 10, 30, 50, 100, and 400 years was calculated using the same method used for annual total discharge.

The drought runoff coefficient of each occurrence probability for the 44 watersheds was calculated by dividing the annual total discharge by annual precipitation.

## 2.3 Collecting data for controlling factors

I assessed 11 indicators, classified into three categories (geological, land use, and topographic factors), as controlling
factors of the drought runoff coefficient.

As a geological factor, I focused on surface geology. I classified the surface geology into four groups (volcanic rock, plutonic rock, metamorphic rock, and sedimentary rock) based on geological creation processes using a subsurface geological map with a scale of 1:200,000 (http://nrb-www.mlit.go.jp/kokjo/inspect/landclassification/download/). The ratio of each surface geology was calculated using a geographic information system (GIS). In addition, metamorphic rock was excluded from the analysis
because the composition ratio was less than 5% in all target watersheds.

The land use data were obtained from the National Survey on the Natural Environment by the Japan Ministry of Environment (http://www.vegetation.biodic.go.jp/legend.html). Classification of land use was based on five categories (coniferous forest, broadleaf forest, mixed coniferous–broadleaf forest, cropland, and urban areas); each class was considered to have different effects on runoff phenomena. The ratio of each land use for the 44 watersheds was calculated by GIS.

I calculated the inverse of channel slope and topographical gradient, form ratio, and roundness as topographic factors. Channel slope was defined as the division of the difference in elevation between the observation station and the headwater by the length of the stream channel. The form ratio was calculated by dividing the watershed area by the square of the length of the stream channel (Horton, 1932). The form ratio approaches 1.0 if the shape of the basin is almost square or circular. The roundness was calculated as the division of the circumference of the same area of a watershed by the boundary length of the
watershed (Miller, 1953). Topography data were obtained from Global 3D Map Service (ALOS World 3D-30 m).

## 2.4 Statistical analysis

To investigate the characteristics of the drought runoff coefficient and its relationship with the controlling factors, an analysis by non-metric multi-dimensional scaling (NMDS) (Kruskal, 1964) was conducted. NMDS refers to a family of related ordination techniques, all of which use rank order information in a (dis)similarities matrix (Coxon, 1982; Gauch, 1982;
Whittaker, 1987). Similarity in the drought runoff coefficient between watersheds was calculated by Bray–Curtis similarity (Bray & Curtis, 1957). As a result of the permutation test, controlling factors closely related to the classification of the drought runoff coefficient ($p < 0.01$) were presented as vectors. Of the 11 indicators used as controlling factors, the topographical gradient was excluded from the analysis because of the strong positive correlation ($r > 0.07$) between it and cropland. In addition, to investigate the difference in controlling factors among groups classified by similarity of the drought runoff
coefficient, the controlling factors of each group were analyzed using one-way analysis of variance and the Kruskal–Wallis





test. Further, Tukey's honestly significant difference (Tukey's HSD) and the Steel–Dwass test were conducted to reveal differences between groups if a significant difference was confirmed among groups.

Next, a generalized linear model (GLM) was developed to formulate a prediction model for the drought runoff coefficient for each occurrence probability. As the explanatory variables, 10 controlling factors were selected, same of the NMDS. The

GLM is an extinction model of a linear model, which allows the incorporation of non-normal distributions of the response variables and linear transformations of the dependent variables (McCullagh and Nelder, 1989). I compared the obtained Akaike information criteria (AIC) (Burnham & Anderson, 2002) of each model by increasing and decreasing the variables. Finally, I adopted the lowest AIC model as the best model for each of the species. GLM was conducted using MASS (Version 7.3-50).

## 3 Results

**3.1 Annual precipitation and drought water volume for each occurrence probability**

The calculation results for annual precipitation, drought runoff volume, and drought water volume per unit drainage area for each occurrence probability are presented in Table 1. The precipitation amount and drought water volume per unit drainage area tended to be high in southwest Japan and low in north Japan. In addition, the differences in precipitation amount and drought water volume per unit drainage area between observation stations tended to decrease, corresponding with the

increasing occurrence probability. Eight types of probability distribution were selected for the calculation of drought water volume. The probability distributions indicated highest adaptability for Gev, which was selected at 23 stations. LN3Q had the second highest adaptability, being selected at seven stations. In the calculation of the precipitation amount, 10 types of probability distribution were selected. Adaptability followed was in the order: Gev (16 stations) > Gumbel (7 stations) > LN3Q (6 stations).

**3.2 Classification of the drought runoff coefficient and controlling factors**

As a result of seriation and clustering using the drought runoff coefficient for each occurrence probability based on NMDS, the 44 stations were classified into three groups. Furthermore, SR and PR (geological factors) and CF, MCBF, UA, and CL (land use factors) were selected as the controlling factors strongly related to the classification of the drought runoff coefficient based on the permutation test (p < 0.01). The selected controlling factors were placed in the positive direction of the first axis

for PR and UA. MCBF was placed in the negative direction of the first axis. CL was placed in the positive direction of the second axis. CF and SR were placed in the negative direction of the second axis (Fig. 2).

Group A (N = 16) was located in the second and third quadrats, composed of watersheds dominated by a mixed coniferous and broadleaf forest. The watersheds belonging to Group A were also characterized by low ratios of urban area and plutonic rock. Group B (N = 16) was located in the first and fourth quadrats, composed of watersheds dominated by urban area or

cropland. The surface geology of watersheds belonging to Group B was dominated by plutonic rock. Group C (N = 12) was located in the third and fourth quadrats, composed of watersheds characterized by a high ratio of the coniferous forest.





The average value of the drought runoff coefficient for each occurrence probability was large, with Group A > B > C. In addition, the difference in the drought runoff coefficient between occurrence probabilities was smaller in Group A than other groups, exhibiting a little difference between the occurrence probabilities of 2 and 400 years. However, in Group C, the drought

runoff coefficient tended to decrease in accordance with the increasing occurrence probability. In Group B, the change in drought runoff coefficient with occurrence probability indicated behavior intermediate between Groups A and C. Although the drought runoff coefficient decreased up to an occurrence probability of 30 years, it had an almost constant value at occurrence probabilities exceeding 30 years (Fig. 3).

### 3.3 Characteristics of controlling factors in each group

Fig. 4 presents a boxplot of the controlling factors for each group. The bold line in the center of the boxplot depicts the median of the data. The top and bottom of the box indicate the third and first quartiles, respectively. In addition, the line located at the top of the box indicates the largest value over than the value, calculated by (first quartile − 1.5 × (third quartile − first quartile)). The line located at the bottom of the box indicates the smallest value less than the value, calculated by (third quartile − 1.5 × (third quartile − first quartile)).

Geological factors VR and SR had similar results. The highest values for both indicators were observed in Group A, followed by those in Group C and Group B. One-way analysis of variance indicated a significant difference among the three groups (p < 0.01). The Tukey's HSD test revealed a significant difference between Group B and the other two groups (p < 0.01) for both factors. However, PR had an opposite trend. The average value for PR was highest in Group B (41%), followed by those in Group C (7.2%) and Group A (2.7%). Results of the Kruskal–Wallis test revealed significant differences among

the groups (p < 0.01). In addition, the Steel–Dwass test revealed that the PR of Group B was significantly higher than that of Group A (p < 0.01) and Group C (p < 0.01).

    As for the land use factors, MCBF was only confirmed in the watersheds belonging to Group A. The average value for UA was highest in Group B (12%), followed by those in Group C (6.4%) and Group A (2.9%). Results of the Kruskal–Wallis test revealed significant differences among the groups (p < 0.01). In addition, the Steel–Dwass test revealed that the UA of Group

A was significantly lower than that of Group B (p < 0.01) and Group C (p < 0.05).

    By contrast, one-way analysis of variance and the Kruskal–Wallis test indicated no significant difference for the land use factors BF, CF, and CL and all of the topographical factors.

### 3.4 Relationship between the drought runoff coefficient for each occurrence probability and the controlling factor

    Analysis of the relationship between the drought runoff coefficient for each occurrence probability and the controlling

factor by the GLM revealed that PR, SR, CF, and CL are decreasing factors for the drought runoff coefficient, whereas BF, MCBF, and Gr are increasing factors. The influence of these controlling factors on the drought runoff coefficient differed among the occurrence probabilities. Indicators of the forest classification were selected as controlling factors for occurrence probabilities of 2 and 10 years, whereas the land use factor, CF, and UA were selected as decreasing factors for the drought





runoff coefficient for occurrence probabilities of over 30 years. Goodness of fit was highest for the occurrence probability of
50 years ($R^2$ = 0.444), and it was lowest for the occurrence probability of two years ($R^2$ = 0.377). VR (geological factor) and
FR and CLR (topographical factors) were not selected as controlling factors of the drought runoff coefficient (Table 2).

## 4 Discussion

### 4.1 Difference in drought runoff coefficient between areas

As a result of drought runoff coefficient classification, observation stations belonging to the Japanese archipelago were
classified into three groups: Groups A, B, and C. The drought runoff coefficient of Group A was characterized by high values
regardless of changes in occurrence probability. However, the drought runoff coefficient of Group C decreased with increasing
occurrence probability. The change in the drought runoff coefficient with increasing occurrence probability for Group B had
an intermediate trend between Groups A and C. The stable and high drought runoff coefficient of Group A, which was
composed of watersheds belonging to the heavy snow area, can be attributed to its more particular precipitation pattern
compared to those of other areas. Takahashi et al. (1978) investigated the drought water volume of this water source area and
explained that the large drought water volume of north Japan results from the stable water supply induced by spring snowmelt
runoff and intermittent rainwater in fall. This water supply contributes to maintaining the groundwater in the drought season.
In addition, the drought risk of the area influenced by spring snowmelt runoff will increase owing to the decreasing
precipitation amounts in winter and spring as caused by climate change. This suggests the importance of snowmelt runoff to
water resource recharge (Wada et al., 2005).

A trend of decreasing drought runoff coefficient with increasing occurrence probability was found in Group C, which is
composed of the southwest Japanese archipelago. In these watersheds, the precipitation amount largely depends on the
concentrated rainfall of a typhoon or a rainy season (Arao & Kaneko, 1985). Therefore, the low supply of water into the ground
during drought results in a low drought runoff coefficient in the case of a high occurrence probability. In addition to the
precipitation pattern, the geology of the watersheds belonging to Group C also seemed to influence the low drought runoff
coefficient. Group C was composed of watersheds with a high ratio of sedimentary rock (Figure 4). Further, the geological age
of the sedimentary rock of these watersheds (the Mesozoic and Paleozoic age) is older than that in other areas (Sudo, 2006).
The low drought runoff coefficient was thought to be caused by the high agglomeration degree of the rock, which is a result
of the high geological age influencing the deep percolation of precipitation.

### 4.2 Controlling factors and the drought runoff coefficient

#### 4.2.1 Occurrence probability of drought and controlling factors

Based on the GLM, which investigated the relationship between the drought runoff coefficient and controlling factors,
geological factors and land use factors (vegetation) influenced the drought runoff coefficient in high-frequency drought,





whereas land use factors and topographic factors were selected as influencing factors in low-frequency drought. This is
considered to be due to the fact that the runoff components that control flow discharge differ depending on drought frequency.
In the case of high-frequency drought, the factors closely related to surface runoff or subsurface flow were selected, and factors
related to the water-table stream with a larger time scale seemed to be selected for low-frequency drought. Previous research
investigating the relationship between flood discharge and controlling factors for multiple occurrence probabilities revealed
that a coniferous forest increases discharge in low-frequency floods, whereas topographical factors increase discharge in high-
frequency floods (Itsukushima et al., 2016). In addition, it is reported that the controlling factor for stream discharge changes
from rainfall to geological factors with threshold of ordinary water discharge (Mushiake et al., 1981). From these research
results, it is clear that the controlling factors change according to the frequency of both flood and drought events.

### 4.2.2 Geological factors and the drought runoff coefficient

Some research has revealed that geology is one of the controlling factors of flow regime (Peters et al., 2003, 2005; Salinas
et al., 2013). The reasons for differences in drought runoff or base flow as a result of geology are that (i) the retention capacity
of groundwater differs based on geology, and (ii) the infiltration capacity of soils differs based on geology (Lacey & Grayson,
1998; Bloomfield et al., 2009). From the GLM, PR and SR (among the geological factors) were selected as controlling factors
that decrease the drought runoff coefficient in high-frequency drought (Table 2). This result is incompatible with the research
result of Mushiake et al. (1981), who noted that granite (classified as a plutonic rock) is a factor in increasing drought discharge.
This difference was caused by the location of the study area and the observation period of the data. Mushiake et al. (1981)
used the average drought value based on a relatively short-term period. In addition, their research focused on a mountainous
river, the drought discharge of which was dominated by surface runoff or subsurface flow. By contrast, Yokoo & Oki (2010)
revealed that geological age has a relation with drought runoff; in particular, based on an investigation of watersheds with an
area of more than 100 km2, quaternary geology was found to be an increasing factor for drought runoff. Therefore, it is
necessary for one to consider both geology type and geological age as indicators when one predicts drought runoff.

In addition to plutonic rock, sedimentary rock was selected as a decreasing factor for the drought runoff coefficient for
occurrence probabilities of 2 and 10 years. The infiltration capacity of sedimentary rock seems to be changed by the degree of
agglomeration. However, flysch (classified as sedimentary rock) has been revealed as a factor for increasing drought or flood
(Gaál et al., 2012). The GLM results support the finding that the low permeability of sedimentary rock is a controlling factor
in high-frequency drought.

While much research has revealed the relationship between geology and drought discharge, some researchers have claimed
a stronger influence of topography than that of surface geology on groundwater level (Condon & Maxwell, 2015). To clarify
the more precise influence of geology, it is important to analyze the relationship between drought and geology under the same
conditions of watershed area, topography, land use, and drought magnitude. In addition, the agglomeration degree of the rock
is closely related to runoff phenomena, as mentioned above. Further research is needed to quantify the relationship between
drought runoff discharge and geology in various regions.





### 4.2.3 Land use factors and the drought runoff coefficient

Changes in the number of available water resources due to an alteration of the rainfall–runoff relationship caused by vegetation changes have long been recognized (Andréassian, 2004). In addition, runoff volume differs between a coniferous

forest and a broadleaf forest owing to dissimilarities in evapotranspiration (Calder 1990; Zhang et al., 2001; Hirano et al., 2009). My research results also indicate the different functions of the coniferous forest and the broadleaf forest. Based on the GLM, the broadleaf forest is selected as an increasing factor for the drought runoff coefficient for high-frequency drought, whereas the coniferous forest is a decreasing factor for low-frequency drought (Table 2). This is thought to be due to the difference in evapotranspiration. Previous research has indicated that the change in runoff volume is larger for a coniferous

forest when a coniferous forest and a broadleaf forest are cleared (Bosch & Hewlett, 1982). Further, the drought runoff volume increases due to the clearing of the coniferous forest (Andreassin, 2004; Brown et al., 2005; Maita & Suzuki, 2007, 2008). These research results support the results of the GLM. Moreover, I presume that the reason for the coniferous forest decreasing the drought coefficient in low-frequency drought is as follows: Since evapotranspiration and canopy interception occur constantly regardless of drought magnitude, the amount of precipitation for surface runoff decreases as the precipitation

amount decreases, and the effects of coniferous forests become dominant. By contrast, the evapotranspiration amount and the runoff volume are altered by the management condition of the forest, the condition of the forest floor, and tree age (Scott & Lesch, 1997; Sakai et al., 2009; Rasoulzadeh & Homapoor Ghoorabjiri, 2014). This research examined the relationship between the runoff coefficient and vegetation type as land use factors for relatively large watersheds. Therefore, the difference between broadleaf and coniferous forests became clear. However, it should be noted that the runoff coefficient could change

even within the same forest type if the targeted watershed is smaller.

Moreover, land use change significantly alters runoff mechanisms (Fohrer et al., 2001). Among land use changes, urbanization increases flood peak discharge (Brown et al., 2009) and decrease the minimum flow (Poff et al., 2006). The main cause of urbanization decreasing the minimum flow is a decrease in the infiltration area and a decline in the base flow due to the consolidation of pipe systems (Simmons & Reynolds, 1982; Leopold, 1968). The GLM results indicate that urban areas

are a decreasing factor for the drought runoff coefficient in low-frequency drought. The composition of tree species in the forest is an important controlling factor for high-frequency drought because the source of surface water mainly depends on rainfall in the upstream area. Therefore, the impact of urbanization is assumed to be relatively low in high-frequency drought. By contrast, the surface water from the upstream area is decreased in low-frequency drought; therefore, the influence of urbanization, such as limitation of rainfall infiltration or supply of surface water from groundwater, is assumed to be dominant.

### 4.2.4 Topographic factors and the drought runoff coefficient


For the relationship between topographic factors and drought runoff, river length, watershed gradient, average watershed width, and altitude were studied as topographic factors influencing base flow (Yokoo & Oki, 2010; Moliere et al., 2009; Engeland & Hisdal, 2009; Castellarin et al., 2004; Abebe & Foerch, 2006). The GLM indicated that channel slope is an



increasing factor for the drought runoff coefficient at occurrence probabilities of 10 years or more (Table 2). This result
supports the research of Moliere et al. (2009) who revealed that zero flow days increase in high-gradient rivers. However, the
topographic factors were not selected as controlling factors for the drought runoff coefficient at an occurrence probability of 2
years. Runoff discharge in high-frequency drought is mainly governed by surface runoff. Therefore, the geological or land use
factors closely related to surface runoff were dominant, rather than topographical factors. On the other hand, the ratio of
groundwater seemed to increase with river discharge in low-frequency drought. Therefore, the topographic factor most closely
related to the groundwater is thought to be selected. Moreover, this study focused on observation stations with various basin
areas, including both mountainous regions and alluvial areas. Interaction between groundwater and surface water is considered
to be more active in alluvial channels; therefore, the drought runoff coefficient is higher in the low-gradient watershed.

## 5 Conclusions

This manuscript reports a first attempt at revealing the relationship between drought runoff and controlling factors
(geological, land use, and topographical factors) in relation to drought magnitude.

Classification results of the drought runoff coefficient across multiple drought magnitudes indicated three types of behavior
for the drought runoff coefficient. The group with watersheds influenced by snowmelt runoff had a high drought runoff
coefficient regardless of drought magnitude. However, the drought runoff coefficient of the group influenced by rainfall
intensity decreased with increasing drought magnitude. The drought runoff coefficient of the remaining group had intermediate
behavior between the aforementioned two groups. In addition, this classification result indicated a significant relationship
between the ratio of plutonic rock, sedimentary rock (geological factors), urban areas, and a mixed coniferous–broadleaved
forest (land use factors).

The GLM revealed that the controlling factor differs depending on drought magnitude. In high-frequency drought, the
drought runoff coefficient was influenced by geological and vegetation factors, whereas land use and topographical factors
influenced the drought runoff coefficient in low-frequency drought. These differences were caused by the differences in the
runoff component, which dominates stream discharge in relation to drought magnitude.

This research clarified that a change in the drought runoff coefficient due to occurrence probability differs depending on
precipitation pattern or climatic zone, and the controlling factors of the drought runoff coefficient changed in accordance with
occurrence probability. Therefore, for effective water resource management, estimation of the drought runoff volume needs to
consider precipitation pattern, geology, land use, and topography. Since the results clarify the controlling factors of drought
runoff for each occurrence probability, this study contributes to effective water resource management by estimating the drought
volume for climatic zones and by predicting changes in drought volume due to climate change. Further research is needed to
investigate applicable climate zones and the influence of catchment scale on the relationship between drought and the
controlling factors.






**Competing Interests**

The author declares that he has no conflicts of interest.

**Acknowledgements**

This work was supported by JSPS KAKENHI grant number JP19H02250.

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







Figure 1: Location of the study site. The 44 observation stations in the Japanese archipelago were considered.




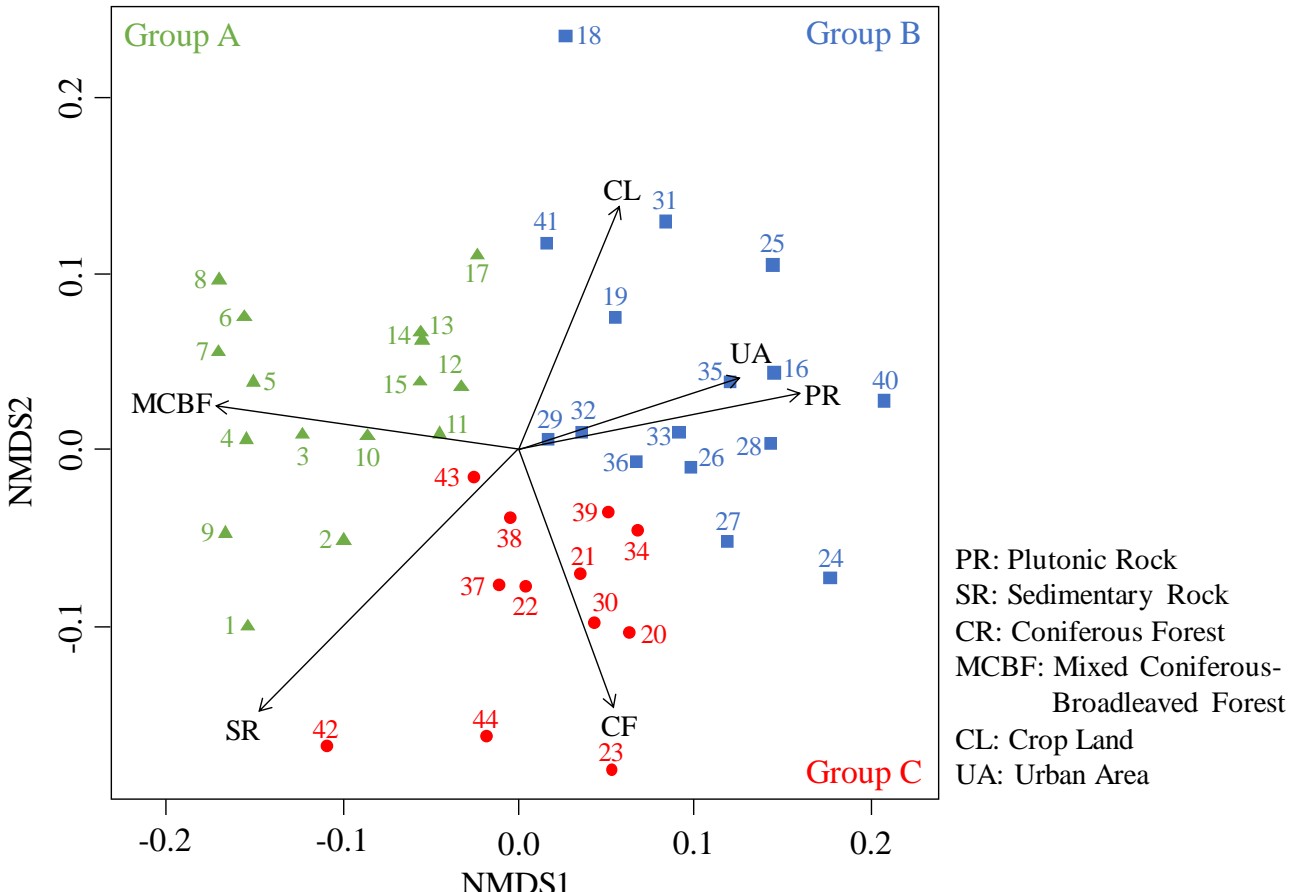

**Figure 2: Results of NMDS using the drought runoff coefficient for each occurrence probability. NMDS: non-metric multi-dimensional scaling**



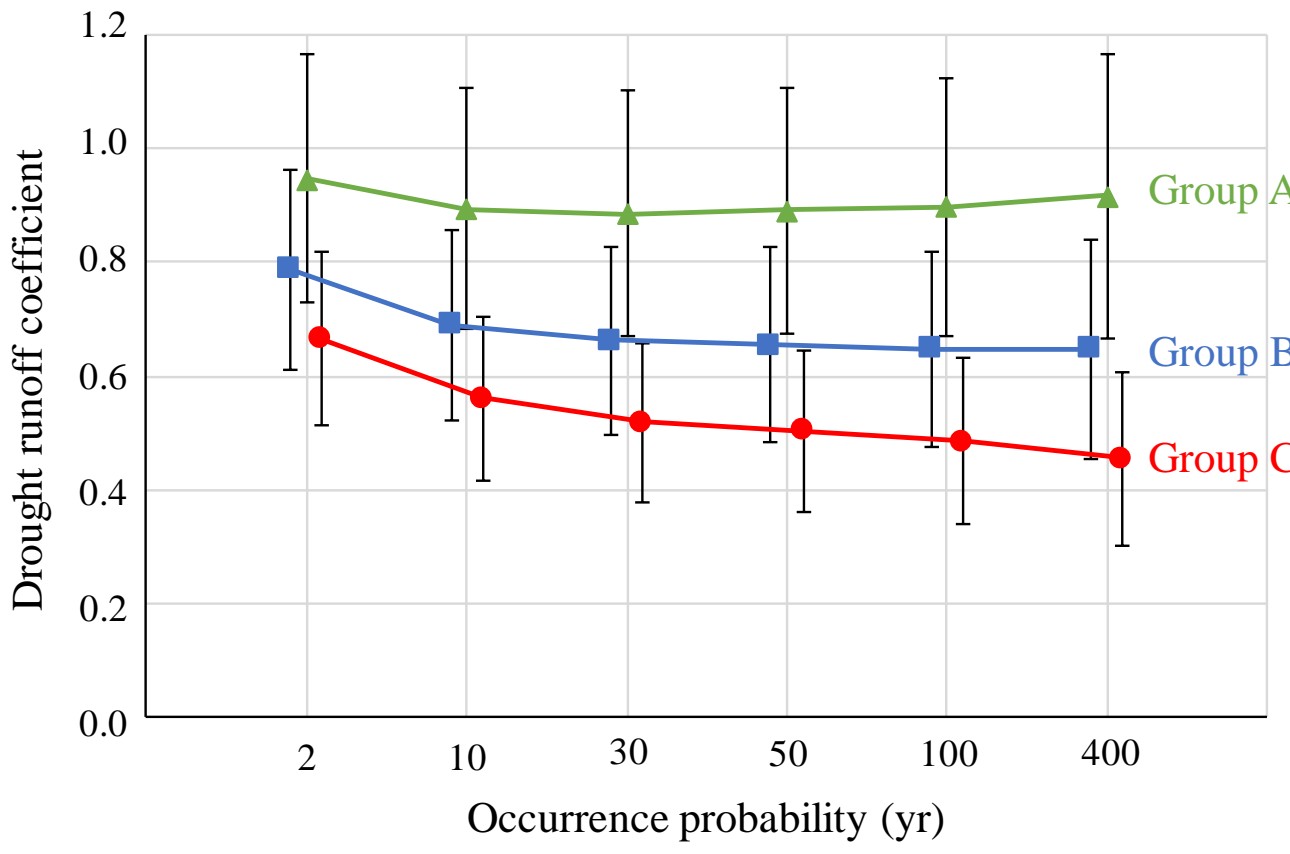


**Figure 3: Average value of the drought runoff coefficient for each occurrence probability across three groups**

**Figure 4: Comparison of controlling factors between groups: (a) VR: volcanic rock, (b) PR: plutonic rock, (c) SR: sedimentary rock, (d) CF: coniferous forest, (e) BF: broadleaf forest, (f) MCBF: mixed coniferous–broadleaved forest, (g) CL: crop land, (h) UA: urban area, (i) CS: channel slope, (j) TGr: topographical gradient, (k) FR: form ratio, (l) Ro: roundness**





**Table 1: The calculation result of annual precipitation, drought runoff volume, and drought water volume per unit drainage area**

| No | Observation station | Basin area (km²) | Precipitation amount for each occurrence probability (mm) | | | | | | N | Model |
|---|---|---|---|---|---|---|---|---|---|---|
| | | | 2 | 10 | 30 | 50 | 100 | 400 | | |
| 1 | Bihoro | 824 | 925 | 731 | 659 | 633 | 602 | 554 | 31 | Gev |
| 2 | Kitami | 1,394 | 794 | 601 | 515 | 481 | 439 | 370 | 31 | Gev |
| 3 | Kaisei | 1,335 | 832 | 646 | 569 | 539 | 504 | 445 | 31 | Gev |
| 4 | Kamishokotsu | 1,051 | 929 | 701 | 595 | 552 | 500 | 411 | 31 | Gev |
| 5 | Makunbetsu | 695 | 961 | 781 | 702 | 672 | 634 | 570 | 31 | Gumbel |
| 6 | Ponpira | 4,029 | 1,161 | 964 | 864 | 822 | 769 | 673 | 31 | Gev |
| 7 | Uryuubashi | 1,661 | 1,439 | 1,199 | 1,085 | 1,038 | 980 | 880 | 31 | Gev |
| 8 | Nakoma | 1,402 | 1,245 | 1,035 | 936 | 897 | 847 | 762 | 31 | LN3Q |
| 9 | Mukawa | 1,228 | 1,192 | 907 | 767 | 711 | 641 | 520 | 31 | Gev |
| 10 | Moiwa | 8,208 | 1,041 | 813 | 709 | 668 | 617 | 532 | 31 | LN3Q |
| 11 | Takanosu | 2,109 | 1,595 | 1,285 | 1,164 | 1,120 | 1,067 | 982 | 31 | Gev |
| 12 | Tsubakikawa | 4,305 | 1,957 | 1,642 | 1,522 | 1,477 | 1,422 | 1,332 | 31 | LN2LM |
| 13 | Todorokibashi | 937 | 2,155 | 1,815 | 1,684 | 1,634 | 1,575 | 1,477 | 31 | LN2LM |
| 14 | Sanbongibashi | 551 | 1,387 | 1,147 | 1,054 | 1,018 | 977 | 906 | 31 | Iwai |
| 15 | Teratsu | 661 | 1,195 | 972 | 890 | 861 | 826 | 772 | 31 | Gev |
| 16 | Kodaiji | 180 | 1,199 | 913 | 793 | 747 | 691 | 598 | 31 | Gev |
| 17 | Shirakawa | 172 | 1,957 | 1,447 | 1,200 | 1,101 | 978 | 772 | 31 | Gev |
| 18 | Kurogo | 580 | 977 | 774 | 685 | 651 | 608 | 536 | 31 | LN3Q |
| 19 | Otome | 760 | 1,381 | 1,117 | 988 | 935 | 870 | 756 | 31 | Gev |
| 20 | Nakazato | 205 | 1,629 | 1,285 | 1,103 | 1,026 | 929 | 755 | 31 | Gev |
| 21 | Takatsudo | 472 | 1,684 | 1,323 | 1,149 | 1,080 | 994 | 847 | 31 | LN3Q |
| 22 | Iwahana | 1,228 | 1,282 | 970 | 845 | 799 | 743 | 653 | 31 | Gumbel |
| 23 | Kitamatsuno | 3,540 | 1,488 | 1,107 | 973 | 923 | 865 | 772 | 31 | Iwai |
| 24 | Iwakura | 501 | 1,634 | 1,266 | 1,117 | 1,060 | 992 | 877 | 31 | Iwai |
| 25 | Hota | 163 | 1,337 | 907 | 717 | 646 | 564 | 434 | 31 | LN3Q |
| 26 | Banjou | 105 | 1,372 | 1,037 | 862 | 790 | 700 | 544 | 31 | Gev |
| 27 | Kashiwara | 962 | 1,397 | 1,035 | 866 | 800 | 720 | 589 | 31 | LN3Q |
| 28 | Hirohara | 195 | 1,869 | 1,616 | 1,520 | 1,484 | 1,443 | 1,381 | 31 | Gev |
| 29 | Huichiba | 837 | 1,721 | 1,412 | 1,287 | 1,239 | 1,181 | 1,082 | 31 | LogP3 |
| 30 | Mitani | 1,049 | 1,818 | 1,435 | 1,307 | 1,261 | 1,208 | 1,122 | 31 | LP3Rs |
| 31 | Otsu | 911 | 1,866 | 1,536 | 1,401 | 1,348 | 1,282 | 1,172 | 31 | LogP3 |
| 32 | Miyatabashi | 123 | 1,475 | 1,112 | 950 | 886 | 808 | 677 | 31 | Gev |
| 33 | Natsuyoshi | 47 | 1,890 | 1,404 | 1,178 | 1,087 | 977 | 791 | 31 | Gev |
| 34 | Nakashima | 326 | 2,262 | 1,664 | 1,379 | 1,264 | 1,125 | 891 | 31 | Gev |
| 35 | Akimatsubashi | 113 | 1,835 | 1,368 | 1,144 | 1,055 | 944 | 756 | 31 | Gev |
| 36 | Hinodebashi | 695 | 1,751 | 1,309 | 1,101 | 1,018 | 917 | 745 | 31 | Gev |
| 37 | Tokusuebashi | 71 | 2,252 | 1,531 | 1,256 | 1,159 | 1,049 | 883 | 31 | Exp |
| 38 | Kawanishibashi | 120 | 2,252 | 1,618 | 1,323 | 1,205 | 1,063 | 824 | 31 | Gev |
| 39 | Myokenbashi | 95 | 1,825 | 1,342 | 1,115 | 1,024 | 912 | 725 | 31 | Gev |
| 40 | Ikemori | 231 | 1,748 | 1,271 | 1,037 | 943 | 826 | 632 | 31 | Gev |
| 41 | Tateno | 386 | 2,688 | 1,992 | 1,727 | 1,629 | 1,511 | 1,316 | 31 | LogP3 |
| 42 | Itsukimiyazono | 227 | 2,217 | 1,639 | 1,414 | 1,330 | 1,232 | 1,068 | 31 | LN3Q |
| 43 | Shiratakibashi | 1,381 | 1,942 | 1,441 | 1,247 | 1,174 | 1,087 | 943 | 31 | LogP3 |
| 44 | Banjyoubashi | 278 | 2,165 | 1,548 | 1,321 | 1,238 | 1,142 | 989 | 31 | Gumbel |



| No | Drought water volume for each occurrence probability ($10^6$m$^3$) | | | | | | N | Model | Drought water volume per unit drainage area for each occurrence probability | | | | | |
|---|---|---|---|---|---|---|---|---|---|---|---|---|---|---|
| | 2 | 10 | 30 | 50 | 100 | 400 | | | 2 | 10 | 30 | 50 | 100 | 400 |
| 1 | 435 | 299 | 254 | 240 | 222 | 196 | 60 | Gev | 0.53 | 0.36 | 0.31 | 0.29 | 0.27 | 0.24 |
| 2 | 699 | 521 | 461 | 441 | 415 | 373 | 60 | LN3PM | 0.50 | 0.37 | 0.33 | 0.32 | 0.30 | 0.27 |
| 3 | 962 | 709 | 637 | 613 | 585 | 541 | 52 | LogP3 | 0.72 | 0.53 | 0.48 | 0.46 | 0.44 | 0.40 |
| 4 | 909 | 680 | 610 | 588 | 565 | 532 | 60 | Gev | 0.86 | 0.65 | 0.58 | 0.56 | 0.54 | 0.51 |
| 5 | 833 | 588 | 500 | 476 | 435 | 385 | 49 | LN3Q | 1.20 | 0.85 | 0.72 | 0.69 | 0.63 | 0.55 |
| 6 | 5,882 | 4,762 | 4,348 | 4,167 | 4,000 | 3,704 | 46 | Gev | 1.46 | 1.18 | 1.08 | 1.03 | 0.99 | 0.92 |
| 7 | 2,326 | 1,852 | 1,667 | 1,587 | 1,515 | 1,370 | 40 | LN3Q | 1.40 | 1.11 | 1.00 | 0.96 | 0.91 | 0.82 |
| 8 | 2,082 | 1,724 | 1,613 | 1,563 | 1,515 | 1,449 | 52 | Gev | 1.49 | 1.23 | 1.15 | 1.11 | 1.08 | 1.03 |
| 9 | 1,250 | 833 | 667 | 625 | 556 | 438 | 42 | LogP3 | 1.02 | 0.68 | 0.54 | 0.51 | 0.45 | 0.35 |
| 10 | 7,143 | 5,263 | 4,545 | 4,348 | 4,000 | 3,448 | 47 | LogP3 | 0.87 | 0.64 | 0.55 | 0.53 | 0.49 | 0.42 |
| 11 | 3,226 | 2,632 | 2,381 | 2,283 | 2,174 | 2,000 | 55 | Iwai | 1.53 | 1.25 | 1.13 | 1.08 | 1.03 | 0.95 |
| 12 | 8,333 | 6,667 | 5,882 | 5,882 | 5,556 | 5,263 | 71 | Gev | 2.07 | 1.65 | 1.46 | 1.46 | 1.38 | 1.30 |
| 13 | 1,961 | 1,538 | 1,389 | 1,333 | 1,266 | 1,149 | 43 | LN3PM | 2.09 | 1.64 | 1.48 | 1.42 | 1.35 | 1.23 |
| 14 | 877 | 676 | 610 | 585 | 559 | 513 | 41 | Iwai | 1.59 | 1.23 | 1.11 | 1.06 | 1.01 | 0.93 |
| 15 | 833 | 625 | 526 | 500 | 476 | 400 | 42 | Gumbel | 1.26 | 0.95 | 0.80 | 0.76 | 0.72 | 0.61 |
| 16 | 137 | 95 | 83 | 78 | 72 | 65 | 36 | Gev | 0.76 | 0.53 | 0.46 | 0.43 | 0.40 | 0.36 |
| 17 | 196 | 137 | 116 | 110 | 101 | 88 | 48 | Gev | 1.14 | 0.80 | 0.68 | 0.64 | 0.59 | 0.51 |
| 18 | 714 | 526 | 455 | 435 | 400 | 357 | 55 | Gumbel | 1.23 | 0.91 | 0.78 | 0.75 | 0.69 | 0.62 |
| 19 | 1,064 | 690 | 524 | 461 | 388 | 274 | 36 | Gev | 1.40 | 0.91 | 0.69 | 0.61 | 0.51 | 0.36 |
| 20 | 217 | 141 | 110 | 98 | 84 | 62 | 37 | Gev | 1.06 | 0.69 | 0.53 | 0.48 | 0.41 | 0.30 |
| 21 | 588 | 370 | 286 | 263 | 227 | 182 | 51 | LN3Q | 1.25 | 0.78 | 0.61 | 0.56 | 0.48 | 0.39 |
| 22 | 901 | 592 | 478 | 439 | 392 | 318 | 42 | Gev | 0.73 | 0.48 | 0.39 | 0.36 | 0.32 | 0.26 |
| 23 | 2,174 | 1,163 | 885 | 794 | 699 | 552 | 49 | LN3Q | 0.61 | 0.33 | 0.25 | 0.22 | 0.20 | 0.16 |
| 24 | 500 | 314 | 240 | 214 | 182 | 133 | 42 | Gev | 1.00 | 0.63 | 0.48 | 0.43 | 0.36 | 0.27 |
| 25 | 200 | 118 | 94 | 86 | 78 | 65 | 24 | Gumbel | 1.23 | 0.72 | 0.58 | 0.53 | 0.48 | 0.40 |
| 26 | 102 | 64 | 52 | 48 | 43 | 36 | 29 | Gumbel | 0.97 | 0.61 | 0.49 | 0.46 | 0.41 | 0.35 |
| 27 | 840 | 552 | 446 | 410 | 369 | 308 | 29 | Exp | 0.87 | 0.57 | 0.46 | 0.43 | 0.38 | 0.32 |
| 28 | 278 | 213 | 189 | 180 | 169 | 154 | 35 | Iwai | 1.42 | 1.09 | 0.97 | 0.93 | 0.87 | 0.79 |
| 29 | 1,205 | 943 | 855 | 820 | 781 | 719 | 32 | LP3Rs | 1.44 | 1.13 | 1.02 | 0.98 | 0.93 | 0.86 |
| 30 | 1,282 | 862 | 704 | 649 | 581 | 474 | 28 | Gev | 1.22 | 0.82 | 0.67 | 0.62 | 0.55 | 0.45 |
| 31 | 1,389 | 1,064 | 935 | 885 | 826 | 730 | 23 | Gumbel | 1.52 | 1.17 | 1.03 | 0.97 | 0.91 | 0.80 |
| 32 | 141 | 93 | 79 | 74 | 68 | 60 | 58 | LogP3 | 1.15 | 0.76 | 0.64 | 0.60 | 0.56 | 0.49 |
| 33 | 76 | 46 | 35 | 31 | 26 | 19 | 30 | Gev | 1.61 | 0.98 | 0.74 | 0.65 | 0.55 | 0.39 |
| 34 | 412 | 260 | 207 | 188 | 167 | 134 | 62 | SqrtEt | 1.26 | 0.80 | 0.64 | 0.58 | 0.51 | 0.41 |
| 35 | 156 | 102 | 84 | 78 | 71 | 60 | 38 | Gumbel | 1.38 | 0.90 | 0.74 | 0.69 | 0.63 | 0.53 |
| 36 | 952 | 595 | 469 | 426 | 376 | 300 | 55 | SqrtEt | 1.37 | 0.86 | 0.68 | 0.61 | 0.54 | 0.43 |
| 37 | 96 | 58 | 45 | 41 | 36 | 28 | 41 | SqrtEt | 1.35 | 0.82 | 0.64 | 0.57 | 0.50 | 0.40 |
| 38 | 185 | 106 | 81 | 71 | 61 | 46 | 38 | Gev | 1.54 | 0.89 | 0.67 | 0.60 | 0.51 | 0.38 |
| 39 | 133 | 78 | 56 | 48 | 38 | 25 | 27 | Gev | 1.40 | 0.82 | 0.59 | 0.50 | 0.40 | 0.26 |
| 40 | 222 | 133 | 109 | 101 | 93 | 80 | 25 | Gev | 0.96 | 0.58 | 0.47 | 0.44 | 0.40 | 0.35 |
| 41 | 714 | 500 | 435 | 400 | 370 | 323 | 24 | Gumbel | 1.85 | 1.30 | 1.13 | 1.04 | 0.96 | 0.84 |
| 42 | 526 | 345 | 278 | 256 | 233 | 192 | 35 | LN3Q | 2.32 | 1.52 | 1.22 | 1.13 | 1.02 | 0.85 |
| 43 | 1,818 | 1,282 | 1,124 | 1,064 | 1,000 | 893 | 66 | LN3PM | 1.32 | 0.93 | 0.81 | 0.77 | 0.72 | 0.65 |
| 44 | 357 | 209 | 165 | 150 | 133 | 108 | 57 | LN3Q | 1.28 | 0.75 | 0.59 | 0.54 | 0.48 | 0.39 |


**Table 2: Analysis of the relationship between drought runoff coefficient of each occurrence probability and controlling factors by GLM**

|  | Occurrence probability | | | | | |
| --- | --- | --- | --- | --- | --- | --- |
|  | 2 | 10 | 30 | 50 | 100 | 400 |
| **Geological factor** | | | | | | |
| VR | | | | | | |
| PR | (−) ** | (−) * | | | | |
| SR | (−) ** | (−) ** | | | | |
| **Land use factor** | | | | | | |
| BF | (+) * | (+) ** | | | | |
| CF | | | (−) ** | (−) *** | (−) ** | (−) *** |
| MCBF | (+) ** | (+) ** | | | | |
| CL | | | | (−) * | | |
| UA | | | (−) * | (−) * | (−) * | (−) * |
| **Topographical factor** | | | | | | |
| CS | | (+) ** | (+) ** | (+) ** | (+) * | (+) * |
| FR | | | | | | |
| RO | | | | | | |
| $R^2$ | 0.377 | 0.441 | 0.435 | 0.444 | 0.421 | 0.430 |
| AIC | −23.013 | −24.676 | −20.005 | −17.291 | −12.615 | −4.9517 |