# Peer review of "Characteristics and controlling factors of the drought runoff coefficient"

_Hydrology and Earth System Sciences, 2019_

## Referee Comment (RC1) · Anonymous Referee #1 · 18 Sep 2019

"general comments" This manuscript by Itsukushima clarified that the controlling factors of drought runoff coefficient depend on drought severity. As is mentioned in the manuscript, many researchers have pointed out the importance of the controlling factors introduced in this manuscript, but the quantitative analysis for them is not enough. Especially, the effective factors for the drought have not been clarified theoretically. I believe this manuscript contributes the effective water resource management. However, the following point is not clear for me. In this manuscript, snowfall and rainfall pattern is considered to be the important factor, but the related factors to them are not included in 11 (or 12) controlling factors. What is the relationship between snowfall or rainfall pattern and the selected controlling factors? I hope this point will be elucidated. Moreover, hypothesis of the mechanism in 4.2.2, which is about geological condition,

is a little bit unclear compared with other sections. I ask you to make discussion more convincing one.

"specific comments" Here are my line-by-line comments: 96 "The drought runoff co-efficient of . . . by annual precipitation" How did you deal with the amount of water withdrawal upstream of the stations? I guess the amount of water withdrawal is not negligible during drought season in some rivers.

110 "topographical gradient" How did you calculate this parameter? Explain the definition by showing the difference from the channel slope.

122 "Of the 11 indicators" & 129 "10 controlling factors" I might misunderstand, but why are they 11 and 10? I thought you selected 12 factors in addition to the metamorphic rock. Then you excluded topographical gradient from them, right? Moreover, you show 12 controlling factors in Figure 4 while you excluded the topographical gradient. It is confusing.

122 "the topographical gradient was excluded" Why did you exclude the gradient instead of cropland? It is reasonable to exclude one of them, but it is necessary to consider the cause of strong correlation. Moreover, in the discussion chapter, it is better to discuss which is the fundamental cause of high or low drought runoff coefficient.

151 "CF and SR were . . . of the second axis" Isn't SR placed in the negative direction of the first axis, too?

186 Does "Gr" mean "TGr" in Figure 4? Unify the abbreviation. Moreover, didn't you exclude the topographical gradient from the analysis?

"4.1 Difference in drought runoff coefficient between areas" In this section, you analyzed the difference of runoff coefficient by focusing on the snowfall and rainfall pattern. You mentioned Group A corresponds to heavy snow area and Group C corresponds to southwest Japan. However, the relationship between these regional characteristics and 11 controlling factors is unclear. It seems that rainfall or snowfall pattern determine
the drought runoff coefficient regardless of 11 controlling factors, but the heavy snow area concentrates in the second and third quadrats where is characterized by low ratios of urban area and plutonic rock as well as high ratios of mixed coniferous-broadleaved forest. Explain the relationship carefully.

211 "Group C was composed of watersheds with a high ratio of sedimentary rock (Figure 4)." How about Group A? It may be related to my comment in line 151, but Group A seems to be composed of watersheds with a high ratio of sedimentary rock in Figure 4.

235 "Mushiake et al. (1981) used the average drought value based on a relatively short-term period." How do you think the average value on a short-term period differs from your values? Introduce the reason why their result is opposite to your result.

239 "Therefore, it is necessary for one to consider both geology type and geological age as indicators" Geological age of sedimentary rock and the difference between quanternary and old volcanic rock are the important factor as Mushiake or Yokoo and Oki showed before. But I do not know the study which addressed the geological age of granite. Explain your hypothesis how the geological age affects the runoff process.

"technical corrections"

Figure 2 "CR" in the legend is considered to be "CF"

Figure 4 & Table 2 Abbreviation of the Roundness is "Ro" in Figure 4 and is "RO" in Table 2. Unify the abbreviation.

191 "FR and CLR . . . were not selected" What is CLR? Isn't it Ro (RO)?
* * *

---

## Author Comment (AC1) · 11 Oct 2019

ãČżGeneral comments This manuscript by Itsukushima clarified that the controlling factors of drought runoff coefficient depend on drought severity. As is mentioned in the manuscript, many researchers have pointed out the importance of the controlling factors introduced in this manuscript, but the quantitative analysis for them is not enough. Especially, the effective factors for the drought have not been clarified theoretically. I believe this manuscript contributes the effective water resource management. However, the following point is not clear for me. In this manuscript, snowfall and rainfall pattern is considered to be the important factor, but the related factors to them are not included in 11 (or 12) controlling factors. What is the relationship between snowfall or rainfall pattern and the selected controlling factors? I hope this point will be elucidated.

Moreover, hypothesis of the mechanism in 4.2.2, which is about geological condition, is a little bit unclear compared with other sections. I ask you to make discussion more convincing one.

Response: I wish to express my strong appreciation to the reviewer for insightful comments on my paper. I feel the comments have helped me significantly improve the paper. I agree with you and have incorporated this suggestion throughout my paper. As you pointed, precipitation is strongly related to the drought runoff volume. Therefore, I used the indicator of drought runoff coefficient which was calculated from dividing the total river runoff by total rainfall in each area. In addition, I added the explanation of the relationship between geological condition and drought runoff coefficient based on your comments.

ãČżSpecific comments 1. Line 96 "The drought runoff co-efficient of . . . by annual precipitation" How did you deal with the amount of water withdrawal upstream of the stations? I guess the amount of water withdrawal is not negligible during drought season in some rivers.

Response: To avoid the effects of artificial flow equalization, I excluded the observation stations whose watershed was subjected to over 10 % occupancy by a dam watershed. This is explained in Line 76. Further, I checked the discharge-duration curve to confirm the no unnatural and extreme reduction of flow discharge.

2. Line 110 "topographical gradient" How did you calculate this parameter? Explain the definition by showing the difference from the channel slope.

Response: According to the suggestion, I added the calculation method of topographical gradient as follow (Line 112-113).

Topographical gradient was obtained by averaging the slope angles calculated by the average maximum method in the watershed (Burrough, 1998).

Additional Reference Burrough, P. A., McDonell, R. A.: Principles of Geographical In-

formation Systems (Oxford University Press, New York), 190, 1998.

3. Line 122 "Of the 11 indicators" & 129 "10 controlling factors" I might misunderstand, but why are they 11 and 10? I thought you selected 12 factors in addition to the meta-morphic rock. Then you excluded topographical gradient from them, right? Moreover, you show 12 controlling factors in Figure 4 while you excluded the topographical gradient. It is confusing.

Response: Initially, I installed 12 indicators as watershed factor. Among the 12 indicators, I delated the indicator of metamorphic rock (composition ratio was less than 5% in all target watersheds) and topographical gradient (strong positive correlation ($r > 0.07$) between it and cropland.). Finally, I used 10 indicators for calculation of NMDS and GLM. Therefore, I corrected the description as follows.

Line 99: I assessed 12 indicators Line 124: Of the indicators Line 131: As the explanatory variables, 10 controlling factors were used,

4. Line 122 "the topographical gradient was excluded" Why did you exclude the gradient instead of cropland? It is reasonable to exclude one of them, but it is necessary to consider the cause of strong correlation. Moreover, in the discussion chapter, it is better to discuss which is the fundamental cause of high or low drought runoff coefficient.

Response: I left the cropland as indicator because cropland is an important parameter that occupies a relatively large area in the target watersheds. In addition, I added the discussion of fundamental cause of drought runoff coefficient as follow (Line 229-233). Further, I added the standard partial regression coefficient of each analysis results in Table 2.

Comparing the standard partial regression coefficient obtained from the GLM according to the occurrence probability, value of MCBF of land use factor was high as that of geological factors in the high-frequency drought. In the drought with occurrence probability of 30 years, the value of land use factor exceeded the geological factor, and

CF was selected as the particularly influential indicator. Further, CS is selected as an important factor in low-frequent drought in addition to CF.

5. Line 151 "CF and SR were . . . of the second axis" Isn't SR placed in the negative direction of the first axis, too?

Response: As requested, I modified the sentence as follow (Line 153- 154).

CF was placed in the negative direction of the second axis and SR was placed in the negative direction of the both of first and second axis (Fig. 2).

6. Line 186 Does "Gr" mean "TGr" in Figure 4? Unify the abbreviation. Moreover, didn't you exclude the topographical gradient from the analysis?

Response: As requested, I modified from TGr to CS (Line 189).

7. "4.1 Difference in drought runoff coefficient between areas" In this section, you analyzed the difference of runoff coefficient by focusing on the snowfall and rainfall pattern. You mentioned Group A corresponds to heavy snow area and Group C corresponds to southwest Japan. However, the relationship between these regional characteristics and 11 controlling factors is unclear. It seems that rainfall or snowfall pattern determine the drought runoff coefficient regardless of 11 controlling factors, but the heavy snow area concentrates in the second and third quadrats where is characterized by low ratios of urban area and plutonic rock as well as high ratios of mixed coniferous-broadleaved forest. Explain the relationship carefully.

Response: In this study, I used the indicator of drought runoff coefficient which was calculated from dividing the total river runoff by total rainfall in each area. However, the drought runoff coefficient was calculated based on the year unit, therefore, the difference tendency of drought runoff coefficient of occurrence probability among groups was caused by the seasonal rainfall pattern with different time scale. Based on your comment, we added the discussion of rainfall pattern as follows (Line 201-206).

In this study, I used the indicator of drought runoff coefficient which was calculated from

dividing the total river runoff by total rainfall in each area. Whereas, the drought runoff coefficient was calculated based on the year unit, therefore, the difference tendency of drought runoff coefficient of occurrence probability among groups was thought to be partly caused by the seasonal rainfall pattern with different time scale. However, it is obvious that the watershed factors have strong influence on the drought runoff coefficient because the characteristic of watershed indicator differs for each classification from the result of NMDS.

8. Line 211 "Group C was composed of watersheds with a high ratio of sedimentary rock (Fig-ure 4)." How about Group A? It may be related to my comment in line 151, but Group A seems to be composed of watersheds with a high ratio of sedimentary rock in Figure 4.

Response: As requested, I added the discussion of comparison with group A as follows (Line 222-224).

Group A is also an area with a large proportion of sedimentary rocks, but it is thought that the difference in geological age and the influence of rainfall patterns are dominant, resulting in a difference in drought outflow rate from Group C.

9. Line 235 "Mushiake et al. (1981) used the average drought value based on a relatively short-term period." How do you think the average value on a short-term period differs from your values? Introduce the reason why their result is opposite to your result.

Response: In accordance with your suggestions, I added the cause of the difference from the results of Mushiake et al. (1981) as follows. (Line 250-253)

In the steep mountain rivers with a small watershed area, rainfall flows out in a short period, and the ratio of surface and intermediate runoff to drought discharge is thought to be larger. In addition, the influence of local deep percolation in bedrock cracks seem to be remarkable in small watershed. Therefore, a certain degree of basin area is

necessary to evaluate the effect of geological factor to drought runoff coefficient.

10. Line 239 "Therefore, it is necessary for one to consider both geology type and geological age as indicators" Geological age of sedimentary rock and the difference between quanternary and old volcanic rock are the important factor as Mushiake or Yokoo and Oki showed before. But I do not know the study which addressed the geological age of granite. Explain your hypothesis how the geological age affects the runoff process.

Response: As requested, I added the hypothesis of the relationship between the geological age and runoff process. (Line 256-258)

Different geological ages differ in the degree of consolidation and result in the difference of degree of deep percolation. Further, as diagenesis progress, water exchange between aquifer and river less likely to occur. Therefore, geological ages is one of the important factor to characterize the drought runoff coefficient.

ãČżTechnical corrections 1. Figure 2 "CR" in the legend is considered to be "CF"

Response: As you pointed, I modified the legend (Figure 2).

2. Figure 4 & Table 2 Abbreviation of the Roundness is "Ro" in Figure 4 and is "RO" in Table 2. Unify the abbreviation.

Response: As you pointed, I unified the abbreviation as RO (Line 550).

3. Line 191 "FR and CLR : : : were not selected" What is CLR? Isn't it Ro (RO)?

Response: As you pointed, I modified to RO (Line 194).

Please also note the supplement to this comment:
https://www.hydrol-earth-syst-sci-discuss.net/hess-2019-330/hess-2019-330-AC1-supplement.pdf

[Figure]

**Supplement:**

[revised manuscript text omitted]
. In this study, I used the indicator of drought runoff coefficient that was calculated by dividing the total river runoff with total rainfall in each area. Whereas, the drought runoff coefficient was calculated based on the year unit, therefore, the difference in the tendency of drought runoff coefficient of occurrence probability among groups was thought to be partly caused by the seasonal rainfall pattern with different time scale. However, it is obvious that the watershed factors have strong influence on the drought runoff coefficient because the characteristic of watershed indicator differs for each classification from the result of NMDS. The stable and high drought runoff coefficient of Group A, which was composed of watersheds belonging to the heavy snow area, can be attributed to its more particular precipitation pattern compared to those of other areas. Takahashi et al. (1978) investigated the drought water volume of this water source area and explained that the large drought water volume of north Japan results from the stable water supply induced by spring snowmelt runoff and intermittent rainwater in fall. This water supply contributes to maintaining the groundwater in the drought season. In addition, the drought risk of the area influenced by spring snowmelt runoff will increase owing to the decreasing precipitation amounts in winter and spring as caused by climate change. This suggests the importance of snowmelt runoff to water resource recharge (Wada et al., 2005).

A trend of decreasing drought runoff coefficient with increasing occurrence probability was found in Group C, which is composed of the southwest Japanese archipelago. In these watersheds, the precipitation amount largely depends on the concentrated rainfall of a typhoon or a rainy season (Arao & Kaneko, 1985). Therefore, the low supply of water into the ground during drought results in a low drought runoff coefficient in the case of a high occurrence probability. In addition to the precipitation pattern, the geology of the watersheds belonging to Group C also seemed to influence the low drought runoff coefficient. Group C was composed of watersheds with a high ratio of sedimentary rock (Figure 4). Further, the geological age of the sedimentary rock of these watersheds (the Mesozoic and Paleozoic age) is older than that in other areas (Sudo, 2006).

The low drought runoff coefficient was thought to be caused by the high agglomeration degree of the rock, which is a result of the high geological age influencing the deep percolation of precipitation. Furthermore, Group A is an area with a large proportion of sedimentary rocks; however, it is thought that the difference in geological age and the influence of rainfall patterns are dominant, thus resulting in a difference in the drought outflow rate from Group C.

**4.2 Controlling factors and the drought runoff coefficient**

**4.2.1 Occurrence probability of drought and controlling factors**

Based on the GLM, which investigated the relationship between the drought runoff coefficient and controlling factors, geological factors and land use factors (vegetation) influenced the drought runoff coefficient in high-frequency drought, whereas land use factors and topographic factors were selected as influencing factors in low-frequency drought. Comparing the standard partial regression coefficient obtained from the GLM according to the occurrence probability, the value of MCBF of land use factor was high as that of geological factors in the high-frequency drought. In the drought with occurrence probability of 30 years, the value of land use factor exceeded the geological factor, and CF was selected as the particularly influential indicator. Further, CS is selected as an important factor in low-frequent drought in addition to CF. This is considered to be due to the fact that the runoff components that control flow discharge differ depending on drought frequency. In the case of high-frequency drought, the factors closely related to surface runoff or subsurface flow were selected, and factors related to the water-table stream with a larger time scale seemed to be selected for low-frequency drought. Previous research investigating the relationship between flood discharge and controlling factors for multiple occurrence probabilities revealed that a coniferous forest increases discharge in low-frequency floods, whereas topographical factors increase discharge in high-frequency floods (Itsukushima et al., 2016). In addition, it is reported that the controlling factor for stream discharge changes from rainfall to geological factors with a threshold of ordinary water discharge (Mushiake et al., 1981). From these research results, it is clear that the controlling factors change according to the frequency of both flood and drought events.

**4.2.2 Geological factors and the drought runoff coefficient**

Some research has revealed that geology is one of the controlling factors of flow regime (Peters et al., 2003, 2005; Salinas et al., 2013). The reasons for differences in drought runoff or base flow as a result of geology are that (i) the retention capacity of groundwater differs based on geology, and (ii) the infiltration capacity of soils differs based on geology (Lacey & Grayson, 1998; Bloomfield et al., 2009). From the GLM, PR and SR (among the geological factors) were selected as controlling factors that decrease the drought runoff coefficient in high-frequency drought (Table 2). This result is incompatible with the research result of Mushiake et al. (1981), who noted that granite (classified as a plutonic rock) is a factor in increasing drought discharge. This difference was caused by the location of the study area and the observation period of the data. Mushiake et al. (1981) used the average drought value based on a relatively short-term period. In the steep mountain rivers with a small watershed area, rainfall flows out in a short period, and the ratio of surface and intermediate runoff to drought discharge is thought to be

larger. In addition, the influence of local deep percolation in bedrock cracks seems to be remarkable in small watershed. Therefore, a certain degree of basin area is necessary to evaluate the effect of geological factor to drought runoff coefficient.

By contrast, Yokoo & Oki (2010) revealed that geological age has a relation with drought runoff; in particular, based on an investigation of watersheds with an area of more than 100 km2, quaternary geology was found to be an increasing factor for drought runoff. Different geological ages differ in the degree of consolidation and result in the difference of degree of deep percolation. Further, as diagenesis progress, water exchange between aquifer and river less likely to occur. Therefore, geological age is one of the important factors to characterize the drought runoff coefficient. 
[revised manuscript text omitted]

[Figure]

1. Bihoro (Abashiri. R)
2. Kitami (Tokoro. R)
3. Kaisei (Yubetsu. R)
4. Kamishokotsu (Shokotsu. R)
5. Makunbetsu (Nayoro. R)
6. Ponpira (Teshio. R)
7. Uryuubashi (Uryuu. R)
8. Nakoma (Shiribetsu. R)
9. Mukawa (Mu. R)
10. Moiwa (Tokachi. R)
11. Takanosu (Yoneshiro. R)
12. Tsubakikawa (Omono. R)
13. Todorokibashi (Koyoshi. R)
14. Sanbongibashi (Naruse. R)
15. Teratsu (Su. R)
16. Kodaiji (Otakine. R)
17. Shirakawa (Abukuma. R)
18. Kurogo (Kokai. R)

19. Otome (Omoi. R)
20. Nakazato (Uzuma. R)
21. Takatsudo (Watarase. R)
22. Iwahana (Karasu. R)
23. Kitamatsuno (Fuji. R)
24. Iwakura (Kizu. R)
25. Hota (Soga. R)
26. Banjou (Saho. R)
27. Kashiwara (Yamato. R)
28. Hirohara (Izushi. R)
29. Huichiba (Maruyama. R)
30. Mitani (Kino. R)
31. Otsu (Hii. R)
32. Miyatabashi (Inunaki. R)

33. Natsuyoshi (Kibe. R)
34. Nakashima (Hikosan. R)
35. Akimatsubashi (Honami. R)
36. Hinodebashi (Onga. R)
37. Tokusuebashi (Tokusue. R)
38. Kawanishibashi (Matsuura. R)
39. Myokenbashi (Ushidu. R)
40. Ikemori (Kase. R)
41. Tateno (Shira. R)
42. Itsukimiyazono (Kuma. R)
43. Shiratakibashi (Ono. R)
44. Banjyoubashi (Banjyou. R)

[revised manuscript text omitted]

---

## Referee Comment (RC2) · Anne Van Loon (Referee) · 18 Nov 2019

The paper by Itsukushima aims to quantify and model the relationship between drought conditions and controlling factors based on geology, land use, and topography. This would have been an interesting topic, but the research does not do what is promised. The analysis used average annual discharge divided by average annual precipitation, which clearly is not the same as drought. Average annual Q/P includes both high- and low-flow periods and the annual timescale is too long for many droughts. This is unfortunately not the only misunderstanding in the paper. Terms are mixed up, a whole body of literature is missed, important factors are left out of the analysis, and the work does not lead to new insights. This paper cannot be accepted for publication in HESS. I explain my rejection below and give more detailed comments in the attached file.

[Figure]

Throughout the paper there is confusion between annual discharge, low flow and drought, both in the literature review and in the analysis. The paper analyses average annual discharge, but talks about drought and low flows. The literature about drought cited in the Introduction is not relevant for the current work on runoff coefficients. Contrary to the claim of the author that this is the first attempt to study the relationship between runoff coefficient and controlling factors, there are already many papers doing this. For example, Berger & Entekhabi (2001), Laaha & Blöschl (2006), Merz & Blöschl (2009), Carey et al. (2010), Sawicz et al. (2011), Ali et al. (2012) and papers by the same and other authors. The probabilistic approach of looking at different return period is maybe new, but the calculation, interpretation and discussion of what this means is unsatisfactory. It is unclear how the runoff coefficient is calculated for each occurrence probability? Have you just divided the 400yr discharge by the 400yr precipitation? Why? What does this mean?

Other methodological flaws include: using different time period of data for precipitation and discharge to compute the runoff coefficient, as the meteorological input might be completely different between these periods, and not taking any climate-related factors into account, as the differences identified in the paper seem (at least partly) to be a function of ET and snow, which are related to latitude and altitude. The analysis should be completely redone and extended. The framework of Wagener et al. (2007) could be a useful guidance.

Finally, the discussion section just mentions a random selection of papers, without thorough synthesis of what the results of this study mean and how they compare to the large body of existing literature on this topic. The discussion and conclusion also contain a lot of misunderstandings and misinterpretations.

[revised manuscript text omitted]

---

## Author Comment (AC2) · 12 Dec 2019

Response to Reviewer

I wish to express my appreciation to the reviewer for insightful comments on my paper. The comments have helped me significantly improve the paper.

General comment: The paper by Itsukushima aims to quantify and model the relationship between drought conditions and controlling factors based on geology, land use, and topography. This would have been an interesting topic, but the research does not do what is promised. The analysis used average annual discharge divided by average annual precipitation, which clearly is not the same as drought. Average annual Q/P includes both high- and low-flow periods and the annual timescale is too long for many

droughts. This is unfortunately not the only misunderstanding in the paper. Terms are mixed up, a whole body of literature is missed, important factors are left out of the analysis, and the work does not lead to new insights. This paper cannot be accepted for publication in HESS. I explain my rejection below and give more detailed comments in the attached file. Throughout the paper there is confusion between annual discharge, low flow and drought, both in the literature review and in the analysis. The paper analyses aver- age annual discharge, but talks about drought and low flows. The literature about drought cited in the Introduction is not relevant for the current work on runoff coeffi- cients. Contrary to the claim of the author that this is the first attempt to study the relationship between runoff coefficient and controlling factors, there are already many papers doing this. For example, Berger & Entekhabi (2001), Laaha & Blöschl (2006), Merz & Blöschl (2009), Carey et al. (2010), Sawicz et al. (2011), Ali et al. (2012) and papers by the same and other authors. The probabilistic approach of looking at different return period is maybe new, but the calculation, interpretation and discussion of what this means is unsatisfactory. It is unclear how the runoff coefficient is calcu- lated for each occurrence probability? Have you just divided the 400yr discharge by the 400yr precipitation? Why? What does this mean? Other methodological flaws include: using different time period of data for precipitation and discharge to compute the runoff coefficient, as the meteorological input might be completely different between these periods, and not taking any climate-related factors into account, as the differences identified in the paper seem (at least partly) to be a function of ET and snow, which are related to latitude and altitude. The analysis should be completely redone and extended. The framework of Wagener et al. (2007) could be a useful guidance. Finally, the discussion section just mentions a random selection of papers, without thorough synthesis of what the results of this study mean and how they compare to the large body of existing literature on this topic. The discussion and conclusion also contain a lot of misunderstandings and misinterpretations.

Response: According to your comments, I have mainly corrected as follows. Details of the correction are described in the specific comments.

ãČžMethod I have clarified the calculation method of drought runoff coefficient of each occurrence probability. Further, I added the specific definition of the low-flow, high and low frequent drought. ãČžResults Based on the definition of the low-flow, high and low frequent drought, I classified the calculation results of runoff coefficient for each occurrence probability into categories of the low-flow, high and low frequent drought. In addition, I added the significant test among three groups for each occurrence probability. ãČžDiscussion Based on the manuscripts introduced by the reviewer, I substantially changed the discussion section to secure the synthesis.

Specific comment: 1. Mention the case study region in the abstract somewhere. The results might be specific to the region.

Response: As requested, I added the explanation of the research region as follow.

Line 10-13 I calculated the drought runoff coefficient for six types of occurrence probability based on past observation data of annual total discharge and precipitation in the Japanese archipelago where multiple climate zones exist.

2. Unclear what you mean with these terms. Are you looking at drought, low flow or minimum flow? What is the drought runoff coefficient? What do you mean with high and low frequency drought?

Response: As pointed out, the definition of low flow, high frequent drought, and low frequent drought was lacked. Therefore, I have defined these terms based on the previous researches as follow. Furthermore, I have classified the calculation results of each occurrence probability based on the definition. Line 106-110 Numerous definitions of hydrological drought have been proposed (Wilhite & Glantz, 1985; Wilhite, 2000). In this study, with reference to Whipple (1966) and Changnon (1980), I defined discharge less than the average annual total discharge as low flow and drought less than the 75% of the average annual total discharge. Furthermore, a discharge of 50%–75% of the average annual total discharge was defined as high-frequency drought, and a discharge of less than 50% was defined as low-frequency drought.

[Figure]

Added references Changnon, S. A.: Removing the confusion over droughts and floods: The interface between scientists and policy makers, Water Int., 5, 10-18, 1980. Whipple, W., Jr.: Regional drought frequency analysis, Proc. ASCE, 92 (IR2) (June), 11–31, 1966. Wilhite, D.A.: Drought as a Natural Hazard: Concepts and Definitions. Drought Mitigation Center Faculty Publications. 69, 2000.

3. Do you mean increase in socio-economic drought related to an increase in demand due to population growth?

Response: This research deals with the future drought risk due to both of population growth and climate change. In this section, I mentioned about the drought due to climate change, therefore, I changed the description as follow.

Line 46-48 However, future prediction of drought aggravation due to climate change and population growth in central Africa (Ahmadalipour et al., 2019) and increasing drought duration and severity in the interior southwest of the United States (Andreadis & Lettenmaier, 2006) have been reported.

4. For clarification, I suggest to add a definition of low flow above, when you define drought.

Response: As pointed out, the definition of low flow, high frequent drought, and low frequent drought was lacked. Therefore, I have defined these terms based on the previous researches as follow. Furthermore, I have classified the calculation results of each occurrence probability based on the definition.

Line 106-110 Numerous definitions of hydrological drought have been proposed (Wilhite & Glantz, 1985; Wilhite, 2000). In this study, with reference to Whipple (1966) and Changnon (1980), I defined discharge less than the average annual total discharge as low flow and drought less than the 75% of the average annual total discharge. Furthermore, a discharge of 50%–75% of the average annual total discharge was defined as high-frequency drought, and a discharge of less than 50% was defined as lowfrequency drought.

5. Add papers by Laaha & Bloschl.

As requested, I added the papers by Laaha & Bloschl as reference.

Line 56-59 For research on flow regime, the factors influencing low flows strongly related to drought have been investigated, through focusing on watershed area, watershed elevation, ratio of urban area or forest cover, and geology (Mushiake et al., 1981; Zecharias & Brutsaert, 1988; Vogel & Kroll, 1992; Laaha & Blöschl, 2005; 2006).

Added references Laaha, G., and Blöschl, G.: 2005. Low flow estimates from short stream flow records - A comparison of methods. J. Hydrol., 306 (1-4), 264-286, 2005. Laaha, G., and Blöschl, G.: A comparison of low flow regionalisation methods-catchment grouping. J. Hydrol., 323 (1-4), 193-214, 2006.

6. What exactly do you mean by "10% occupancy by a dam watershed, is this the area of the reservoir itself or the area of a subcatchment in which a dam is located? Is area the best criterion? How do you know that effects of regulation are negligible when the dam area is less than 10%?

Response: The threshold of 10 % indicates an area of sub-catchment of the dam. There are few watersheds which have abundant discharge data without dam due to active water resource development. Therefore, we used the data of observation stations whose watershed was subject to under 10 % to secure the number of stations. As requested, I added the explanation of dam watershed and getting information of watershed of dams as follow.

Line 75-78 To extract stations where the impact of flow regime regulation due to a dam is small, observation stations whose watershed was subject to over 10% occupancy by the area of a sub-catchment in which a dam is located were excluded (Fig. 1). The information about the sub-catchment areas of dams was obtained from the Japan Dam Foundation (2019).

Added reference Japan Dam Foundation.: Dam year directory, 2019.

7. Why annual? Not low flow. . .

As requested, I added the explanation of the using annual total discharge as the indicator to calculate the drought runoff coefficient as follow.

Line 80-84 Propagation of a precipitation anomaly to the streamflow was explained by the yearly scale (Changnon, 1987; Van Loon, 2015), I used the annual total discharge as the indicator to calculate the drought runoff coefficient. Annual discharge was frequently used as the evaluation indicator of drought (Agnew &Warren, 1996; McMahon & Finlayson, 2003; Henny et al., 2007; Lorenzo-Lacruz et al., 2010).

Added reference Agnew, C., and Warren, A.: A framework for tackling drought and land degradation. Journal of Arid Environments. 33 (3), 309-320, 1996. Changnon S. A.: Detecting drought conditions in Illinois. Illinois State Water Survey Champaign, Circular 169, 1987. Henny, C. J., Hill, E. F., Grove, R. A., and Kaiser, J.L.: Mercury and Drought Along the Lower Carson River, Nevada: I. Snowy Egret and Black-Crowned Night-Heron Annual Exposure to Mercury, 1997–2006. Archives of Environmental Contamination and Toxicology. 53(2), 269-280, 2007. Lorenzo-Lacruz, J., Vicente-Serrano, S. M., López-Moreno, J. I., Beguería, S., García-Ruiz, J. M., and Cuadrat, J.M.: The impact of droughts and water management on various hydrological systems in the headwaters of the Tagus River (central Spain). Journal of Hydrology, 386(1-4), 13-26, 2010. McMahon, T.A., and Finlayson, B.L.: Droughts and anti‐droughts: the low flow hydrology of Australian rivers. Freshwater Biology. 48 (7), 1147-1160, 2003. Van Loon, A.F.: Hydrological drought explained. WIREs Water. 2, 359-392, 2015.

8. Why where these probability distributions used? Why are they suitable for the data? Please add a table of advantages and disadvantages of using these distributions for the calculation of low flow return flows.

Response: As requested, I added the references which explain the advantages and

disadvantages of these hydrological frequency analysis (in line 93-95).

Added reference Etoh, T., Murota, A., and Nakanishi, M.: SQRT-Exponential Type Distribution of Maximum. Hydrologic Frequency Modeling. 253-264, 1987. Griffis, V. W.: Flood Frequency Analysis: Bulletin 17, Regional Information, and Climate Change, Ph. D. Dissertation, Cornell University, 2006. Griffis, V. W., and Stedinger, J. R.: Log-Pearson type 3 distribution and Its application in flood frequency analysis. I: Distribution characteristics. Journal of Hydrologic Engineering. 12(5), 482-491, 2007. Interagency Committee on Water Data.: Guidelines for Determining Flood Flow Frequency: Bulletin 17 B (revised and corrected), Hydrology Subcommittee, Washington, D. C., 1-28, 1982. Ishihara, T., and Takase, N. The logarithmic-normal distribution and its solution based on moment method. Transactions of the Japan Society of Civil Engineers. 47, 18-23, 1957. Stedinger, J.R., Vogel, R. M., and Foufoula-Georgiou.: Frequency analysis of extreme events, Chap. 18, Handbook of Hydrology, (Ed.) Maidment, D. R., McGraw-Hill., NewYork, 1993. Takara, K.: Frequency analysis of hydrological extreme events and how to consider climate change. The Nineteenth IHP training course (International Hydrological Program), 2009. Takara, K., and Tosa, K.: Application of probability distributions with lower and upper bounds to hydrologic frequency analysis. Proceedings of hydraulic engineering, JSCE. 43, 121-126, 1999. Takara, K., and Takasao, T.: Comparison of parameter estimation methods for hydrologic frequency analysis model. Proceedings of hydraulic engineering, JSCE. 34, 7-12, 1990. Yue, S., Ouarda, T. B. M. J., Bobée, B., Legendre, P., and Bruneau, P.: The Gumbel mixed model for flood frequency analysis. Journal of Hydrology. 226(1-2), 88-100, 10.1016/S0022-1694(99)00168-7 1999.

9. For precipitation and discharge? Same 30 year period used between stations and between P and Q?

Response: As for the precipitation amount, same 30 year period was used between stations. Further, the observation period of rainfall and discharge are overlapped.

10. In Table 1 I noticed that you only used 30yr for P, whereas you used various time periods for Q. If you do this, you cannot compare P and Q or calculate a runoff coefficient as they are derived from different time period with different meteorological input.

Response: As requested, I added the explanation of the adequateness of the method. In order to calculate the most probable numerical value among the limited information, the probability distribution that best fits P and Q is adopted. The result of statistical analysis and calculation indicates the adequacy of this method. In addition, previous research revealed that the stability of reproduction statistics increased if the samples are more than about 30. Therefore, I adopted the method.

Line 98-102 Data from observation stations with an observation period of over 30 years were used based on the research result, which indicates that the stability of reproduction statistics increases if the samples are more than about approximately 30 (Takara & Kobayashi, 2009).

Added reference Takara, K., and Kobayashi, K.: Hydraulic analysis methods suitable to the sample size of extreme events. Annual journal of hydraulic engineering, JSCE. 53, 205-210, 2009.

11. Why this method?

Response: As requested, I added the explanation of the reason for the adaptation of this method.

Line 100-102 A sample of the average depth of rainfall over the watershed area was calculated using a Voronoi diagram for objectively considering the area effect of the rainfall at the watersheds.

12. How is this the drought runoff coefficient if you just divide the annual values?

Response: As you have pointed out, the explanation was insufficiently. I modified the calculation of drought runoff coefficient as follow.

Line 104-106 The drought runoff coefficient of each occurrence probability for the 44 watersheds was calculated using the following equation: Qn / (Pn * A) (Qn: estimated total discharge of each occurrence probability, Pn: estimated precipitation amount of each occurrence probability, n = 2, 10, 30, 50, 100, and 400, A: watershed area).

13. Also, from Table 1 it seems that the discharge and precipitation values had different units. This means that you do not get a dimensionless coefficient if you divide them.

Response: I calculated the drought runoff coefficient by dividing total discharge (m3) by catchment area (m2) and rainfall (m). I modified the calculation of drought runoff coefficient as follow.

Line 104-106 The drought runoff coefficient of each occurrence probability for the 44 watersheds was calculated using the following equation: Qn / (Pn * A) (Qn: estimated total discharge of each occurrence probability, Pn: estimated precipitation amount of each occurrence probability, n = 2, 10, 30, 50, 100, and 400, A: watershed area).

14. What do you mean? How did you increase / decrease the variables?

Response: I used the stepwise selection method proposed by Efroymson (1960). I modified the sentence and added a reference as follows.

Line 147-149 I compared the obtained Akaike information criteria (AIC) (Burnham & Anderson, 2002) of each model by the stepwise selection method (Efroymson, 1960).

Added reference Efroymson M. A.: Multiple regression analysis. In: Ralston A, Wilf HS, editors. Mathematical methods for digital computers. New York: Wiley; 1960.

15. Following the methodology you did not calculate drought, only annual streamflow.

Response: As pointed out, the definition of low flow, high frequent drought, and low frequent drought was lacked. Therefore, I have defined these terms based on the previous researches as follow. Furthermore, I have classified the calculation results of each occurrence probability based on the definition.

Line 106-110 Numerous definitions of hydrological drought have been proposed (Wilhite & Glantz, 1985; Wilhite, 2000). In this study, with reference to Whipple (1966) and Changnon (1980), I defined discharge less than the average annual total discharge as low flow and drought less than the 75% of the average annual total discharge. Furthermore, a discharge of 50%–75% of the average annual total discharge was defined as high-frequency drought, and a discharge of less than 50% was defined as low-frequency drought.

Line 161-165 From the calculation of the total discharge of each occurrence probability, the percentage to the average annual discharge was 96%, 67%, 56%, 53%, 48%, and 42% for the occurrence probability of 2, 10, 30, 50, 100, and 400 years, respectively. Therefore, the total discharge of the occurrence probability of 2 years corresponded to the low-flow; 10, 30, and 50 years corresponded to the high-frequency drought; and 100 and 400 years corresponded to the low-frequency drought.

16. Show clustering results.

Response: Clustering results were shown in Figure 2. Further, I have added the explanation of clustering results as follows.

Line 167-168 From seriation and clustering using the drought runoff coefficient for each occurrence probability based on NMDS, the 44 stations were classified into three groups (Group A, B, and C, as shown in Fig. 2).

17. How were these selected?

Response: As you requested, I have added the explanation of the selecting method of variables as follow.

Line 137-138 From the permutation test (n = 999), controlling factors closely related to the classification of the drought runoff coefficient (p < 0.01) were presented as vectors.

18. What do you mean? Refer to Figure 3.

Response: As requested, I modified the sentence as follow.

Line 180-181 To compare the runoff coefficient among groups, the average value of the runoff coefficient was large in order of group A, B, and C in all occurrence probabilities. 19. Also, I do not think that the differences between the groups are statistically significant with this high amount of overlap between the groups. Needs to be tested.

Response: As requested, I have conducted thw significant test among three groups for each occurrence probability. The results were added as follows.

Line 186-189 From the significant test among the three groups for each occurrence probability, a significant difference between groups A and C was confirmed in all occurrence probabilities (p < 0.01). In addition, a significant difference between groups A and B was confirmed in the occurrence probability of 10, 30, 50, 100, and 400 years (p < 0.01).

20. Move to figure caption

Response: As requested, I have moved the sentences to figure caption (Line 665-669).

21. Not surprising. Probably due to low ET?

Response: As pointed, this is due to the low ET. I have added the explanation and references as follow.

Line 234-235 This is also due to the low evapotranspiration in the high-latitude area (Ahn & Tateishi, 1994; Zhang et al., 2011).

Added reference Ahn, C-H., and Tateishi, R.: Development of Global Land Surface Evapotranspiration and Water Balance Data Sets. Journal of the Japan society of photogrammetry and remote sensing. 33, 48-61, 1994. Zhang, K., Kimball, J. S., Kim, Y., and Mcdonald, K. C.: Changing freeze-thaw seasons in northern high latitudes and associated influences on evapotranspiration. Hydrological Processes. 25(26), 4142-4151, 2011.
22. This is not relevant as you only looked at annual Q, not at droughts or seasonality.

Response: Based on your comment, I defined the low flow, drought runoff coefficient as follows.

Line 108-110 I defined discharge less than the average annual total discharge as low flow and drought less than the 75 % of the average annual total discharge. Furthermore, discharge of 50-75% of the average annual total discharge was defined as high-frequent drought, and discharge of less than 50% is defined as low-frequent drought.

23. Again, you are not researching drought.

Response: Based on your comment, I defined the low flow, drought runoff coefficient as follows.

Line 108-110 I defined discharge less than the average annual total discharge as low flow and drought less than the 75 % of the average annual total discharge. Furthermore, discharge of 50-75% of the average annual total discharge was defined as high-frequent drought, and discharge of less than 50% is defined as low-frequent drought.

24. What do you mean?

Response: As requested, I modified the sentences as follows.

Line 264-267 In the total discharge of occurrence probability of 2 and 10 years, geological factors and land use factors were selected as the controlling factors. These factors were closely related to the surface runoff or subsurface flow. In contrast, for the low-frequency drought, factors related to the larger time-scale hydrological cycle, such as ground water level, were apparently selected.

25. There is ample research on controlling factors of the runoff coefficient. You are not researching drought.

Response: As requested, I modified the sentence as follow.

Line 351-352 This manuscript reports the relationship between drought runoff and controlling factors (geological, land use, and topographical factors) in relation to the magnitude of occurrence probability.

26. Nothing new here.

Response: As requested, I modified the sentence as follow.

Line 366-367 Therefore, for effective water resource management, estimation of the drought runoff volume needs to consider precipitation pattern, geology, land use, and topography for corresponding to the magnitude of the drought.

27. Figure 2 How do you mean "for each occurrence probability"? You would get a different number for each probability and so a different figure. Or have you averaged all runoff coefficients for all probabilities?

Response: I used the dataset of runoff coefficient of six occurrence probability (2, 10, 30, 50, 100, and 400 year). I have modified the caption of Figure 2 as follows.

Line 656-657 Figure 2: Results of NMDS using the drought runoff coefficient of six occurrence probabilities (2, 10, 30, 50, 100, and 400 years). NMDS: non-metric multi-dimensional scaling
* * *